# Phage Therapy as a Novel Strategy in the Treatment of Urinary Tract Infections Caused by *E. Coli*

**DOI:** 10.3390/antibiotics9060304

**Published:** 2020-06-05

**Authors:** Beata Zalewska-Piątek, Rafał Piątek

**Affiliations:** Department of Molecular Biotechnology and Microbiology, Chemical Faculty, Gdańsk University of Technology, Narutowicza 11/12, 80-233 Gdańsk, Poland; rafpiate@pg.edu.pl

**Keywords:** urinary tract infections, uropathogenic *E. coli*, antibiotics, alternative treatment, bacteriophages, phage therapy

## Abstract

Urinary tract infections (UTIs) are regarded as one of the most common bacterial infections affecting millions of people, in all age groups, annually in the world. The major causative agent of complicated and uncomplicated UTIs are uropathogenic *E. coli* strains (UPECs). Huge problems with infections of this type are their chronicity and periodic recurrences. Other disadvantages that are associated with UTIs are accompanying complications and high costs of health care, systematically increasing resistance of uropathogens to routinely used antibiotics, as well as biofilm formation by them. This creates the need to develop new approaches for the prevention and treatment of UTIs, among which phage therapy has a dominant potential to eliminate uropathogens within urinary tract. Due to the growing interest in such therapy in the last decade, the bacteriophages (natural, genetically modified, engineered, or combined with antibiotics or disinfectants) represent an innovative antimicrobial alternative and a strategy for managing the resistance of uropathogenic microorganisms and controlling UTIs.

## 1. Introduction

Acute, recurrent, or chronic urinary tract infections (aUTIs, rUTIs, or cUTIs) are the most prevalent microbial diseases, affecting about 150 million people every year worldwide. In the United States, up to 11-million people suffer from UTIs yearly [1,2]. UTIs as the global health problem reducing the quality of life of persons affected by the urinary diseases also generate high social costs (i.e., medical expenses and hospitalizations) [3,4].

Urinary tract infections (UTIs) are considered to be the second most common bacterial diseases after pneumonia, mainly due to the high frequency of period recurrences and chronicity of infections. UTIs are also a leading cause of septicemia [5,6,7]. The widespread and uncontrolled use of antibiotics during therapy and prophylaxis of UTIs results in the continuing increasing antibiotic resistance and emergence of multi-drug resistant (MDR) and extensively drug-resistant (XDR) uropathogens [8,9].

These painful and economically costly infections affect women of all ages (although with different prevalence) more often than men. More than 50% of women experience at least one UTI episode through their lifetime, the consequence of which is usually antibiotic therapy [1,10]. In addition, approximately 20–40% of women who have already experienced an initial UTI will suffer from a recurrence of infection within 3–4 months. The recurrent or chronic infections also often require repeated antibiotic treatment. However, therapeutic regimens do not distinguish between recurrent and new episodes of UTIs, because they are both generally thought to be derived from parenteral sites (e.g., gastrointestinal, vaginal) [11]. For comparison, in the case of young children, one in three affected by a UTI before the first year of life will undergo a recurrent infection within three years, and 18% within a few months [12]. UTIs are also a significant cause of morbidity and mortality in the elderly, accounting for 15.5% of hospitalization and 6.2% of deaths of people aged 65 years or older [6,13].

Classification, Etiologic Factors of UTIs and the Main Pathogenic Mechanisms

When considering the clinical aspect, UTIs are classified as uncomplicated and complicated (Table 1). Uncomplicated UTIs generally affect healthy people who do not show structural, neurological, and physiological abnormalities of the urinary tract. The infections of this type are associated with lower and upper UTIs (urethritis, cystitis, and pyelonephritis, respectively) [4,14,15]. Complicated UTIs occur in individuals with obstructions and foreign bodies in urinary tract (compromising its action), urinary retention due to illness or immunity disorder. In the United States, approximately 70–80% of complicated UTIs are associated with indwelling catheters (i.e., catheter-associated UTIs, CAUTIs) [16,17,18].

UTIs are caused by various uropathogens, including Escherichia coli (~85%), Proteus mirabilis, Klebsiella pneumoniae, Staphylococcus saphrophyticus, Staphylococcus aureus, group B Streptococcus, Enterobacter and Enterococcus species, and Candida spp [4,19].

*Escherichia coli* (EC) from *Enterobacteriaceae* family is a frequent commensal inhabitant of the human or animal gastrointestinal tract that can take pathogenic nature under specific host conditions and virulence factors. The pathogenic *E. coli* strains are differentiated into two types: intestinal/diarrheagenic *E. coli* (InPEC pathotypes) rarely causing disease outside the intestinal tract and extraintestinal *E. coli* (ExPEC pathotypes) with the ability to spread, colonize other niches (e.g., nervous system, urinary tract, blood), and, consequently, resulting in illness (Table 2) [20,21,22].

Among the common pathogens causing urinary diseases, uropathogenic *E. coli* strains (UPECs) are the most dominant etiologic agent for both complicated (50–65%) and uncomplicated (75–85%) UTIs [2,4,19,23]. UPEC isolates are also a very serious problem for pregnant women, constituting 80–90% of bacteria that cause urinary infections in pregnancy [24]. In addition, up to 68% of rUTIs are caused by UPECs that are identical to the original strain, causing the primary infection [25]. Analysis of bladder biopsy cultures that were collected from women with rUTI symptoms after antibiotic treatment showed that 24% of them had positive biopsy bacterial cultures, including sterile urine [26]. It has also been revealed that UPECs persist and reappear in the bladder, despite antibiotic treatment in the mouse model of UTI. This phenomenon is associated with the adhesive and invasive properties of UPEC bacteria that colonize the bladder urothelium. Inside umbrella uroepithelial cells lying on the bladder luminal surface, bacteria are protected from the host’s immune response. A single bacterium can replicate up to 10^4^ or more within hours of invasion, while forming biofilm-like intracellular communities (IBCs) [27,28]. IBCs development is a transient process and it ends at the stage of dispersion, during which the bacteria filament and escape from invaded host cells, and then disperse to neighboring host cells, where the IBC cycle can be repeated [4,29]. The presence of IBCs resembling biofilm development and bacterial filaments was found in the urine of women suffering from acute and recurring infections, but not in healthy control individuals or in the case of infections that are caused by Gram-positive microorganisms (not forming IBCs) [30,31]. The abovementioned uropathogenic cascade, including the invasion of bladder epithelial cells (BECs), intracellular proliferation, escape, and spread of filamentous bacteria from BECs to neighboring cells, was also induced in the human BEC infection model based on a flow chamber culture system [32,33].

UPEC isolates are also able to establish quiescent intracellular reservoirs (QIRs) within deeper uroepithelial transitional cells, where they can exist in the form of isolated, membrane-bound, actin cell compartments. In this way, the host’s immune system is unable to identify them up to several months. Unlike the metabolically active structures of IBCs, QIRs are composed of 4–10 non-replicating bacterial cells that remain viable for many months and they can be activated as a potential source of UTI recurrence (over 30% of bladder infections recur) [4,14,34]. The obtained results indicate that the bladder colonized by persistent bacteria establishing IBCs and QIRs is the source of bacterial dissemination, chronicity, and recurrence of infections. In addition, it has been observed that the expression of UPEC-encoded virulence factors (i.e., iron and zinc acquisition systems, capsule, type 1 fimbriae, microcin secretion genes) can change during the transition of uropathogens from intestinal growth to growth in the urinary tract and also during the course of UTI, as compared to various infected individuals [35,36]. This indicates that the need for a particular pathogenic factor might vary as the urinary infection develops. The analysis of the transcriptional profile of UPECs allowed for distinguishing universal features used by uropathogens that enable them to colonize the urinary tract, survive in such environment, and progress the pathogenic process. Based on the studies performed, a transcription program that was common to genetically differentiated UPEC strains (isolated from patients during uncomplicated UTIs) was revealed. The in vivo gene expression program was characterized by increased regulation of translation and replication machinery, providing a mechanism for the rapid growth of UPECs in affected persons [37].

This creates the need to develop new forms of UTIs or rUTIs diagnosis and treatment, such as intravesical antibiotic administration, vaccines, receptor analogues, pilicides and curlicides, bacterial interference, or phage therapy [2,8,38,39,40]. Currently, in the scientific world, among the above therapeutic approaches, phagotherapy appears to be the most promising alternative to fight various pathogens, including uropathogens that are resistant to commonly available antibiotics. Here, we briefly reviewed the potential of bacteriophages as an innovative strategy and alternative to conventional antibiotics for the treatment of UTIs caused by UPECs.

## 2. Antibiotic Treatment of UTIs Caused by UPECs

The routine treatment of UTIs based on antibiotics is effective in the case of many affected individuals [11,41]. However, chronic and recurrent urinary infections are still an important public health problem. Constant abuse of antibiotics and the practice of describing them in the treatment or prevention of UTIs without bacterial characterization is also a reason for the rapid development of drug resistance patterns of uropathogens causing urinary diseases [42,43].

Currently, trimethoprim (TMP), sulfamethoxazole (SMX), cotrinoxazole (TMP/SMX), and ciprofloxacin, ampicillin, second or third-generation cephalosporins, belonging to the group of β-lactam antibiotics, are the most recommended and frequently used during UTI therapy against UPEC isolates [44,45]. Additionally, carbapenems (a group of β-lactam antibiotics related to penicillins and cephalosporins), like imipenem (100%), ertapenem (99.98%), amikacin (99.94%), and nitrofurantoin (99.91%), are considered the best antibiotics for UPECs, also including β-lactamase (ESBL) strains [46].

Despite the effectiveness of many therapeutic regimens, treatment of UTIs is becoming increasingly difficult due to the global emergence of antibiotic-resistant UPECs and a decrease in the efficacy of oral therapies [42]. Currently, an increase in the resistance rates of UPEC isolates to ampicillin, cotrimoxazole, cephalosporins, amoxicillin, clavulanate and nalidixic acid, cefradine, aminopenicillins, and cefuroxime is observed [46,47,48,49,50]. Some UPECs also show resistance to amikacin and ciprofloxacin [11]. The growing number of TMP-resistant UPECs also limits its use as a single prophylactic factor or in combination with SMX, for women suffering from recurrent UTIs. Furthermore, therapeutic use of fluoroquinolones (as second line drugs, especially for the treatment of pyelonephritis and uncomplicated acute cystitis) is reduced because of the significant increment of resistant UPECs and ecological side effects that are caused by this group of antibacterial chemotherapeutics. It was found that, in European countries, the resistance of UPEC to fluoroquinolones increased to ~50%, whereas, in Vietnam, China, and India, up to 70% [46,51].

The ESBL-EC isolates producing the extended-spectrum of β-lactamases with broad activity against cephalosporins, penicllins, carbapenems, and other antibiotics are a common global therapeutic problem [52,53,54,55]. The ESBLs, plasmid or chromosomally encoded, act by breaking the β-lactam ring of antibiotics and inactivating them [53]. The most worrying are ESBLs that are encoded on plasmids that may harbor other resistance genes conferring the ability of uropathogens to inactivate aminoglycosides, quinolones, and sulfonamides. The presence of such plasmids in bacterial cells results in the emergence and spread of multi-drug resistance among them, which creates the need to develop alternative therapeutic options, such as phage therapy [53,55].

## 3. Bacteriophages—Brief Characteristics

Bacteriophages are viruses that attack bacterial cells and occur everywhere they appear. It can be assumed that there is at least one type of specific phage for each existing bacterial strain. They are characterized by high specificity, usually one virus species can only multiply in one bacterial species [56,57].

The bacterial parasites contain DNA or RNA as the genetic material and can only replicate in the host organism. Phages, which are based on their replication cycles, can be divided into two types as lytic (virulent) phages causing bacterial lysis and lysogenic (temperate) phages associated with the integration of their genetic material into the bacterial genome (as a prophage) and replication with it. Under certain environmental conditions, prophage can activate and initiate the lytic cycle, leading to the degradation of the host cell [58,59,60].

Bacteriophages are widespread in various environments. They are the most numerous and diverse group of organisms on the earth, reaching up to 10^31^ molecules. They can colonize soil, fresh, and salt waters, and even hot springs. They were also isolated from the human or animal organisms (e.g., gastrointestinal system, saliva, urine, feces) [56].

Bacteriophages occur in large quantities in the human body. Many phage niches (e.g., gastrointestinal) are well known. Phages that are part of the human urinary microbiota are poorly understood and studied. Initial viral metagenomic studies of the urinary phages indicated a huge variety of new lytic phage sequences, which far outweigh (>99%) the number of eukaryotic viruses [61]. However, these studies did not include the bladder lysogenic phage populations. Based on recent research, it was shown that the phages dominated within the genome of bladder bacteria. Most of the tested genomes (86%) contained at least one phage sequence. The analyzed phage sequences generally did not reveal homology to the sequences deposited in public databases. In addition, variation in phage number was observed between bacteria that were isolated from asymptomatic healthy woman and those with symptoms of bladder infections. This indicates that phages in the bladder can stabilize bacterial metabolism and community structure. However, further investigations of the bladder phagenome are needed in order to clarify their role in health and disease status during UTI episodes [62].

## 4. History of Bacteriophage Discovery

The first reports indicating the existence of phages come from 1896. Ernst Hankin, an English bacteriologist, revealed in 1896 that the waters of the Ganges and Jamuna rivers in India had the bactericidal activity against *Vibrio cholerae*. The river water that was heated in closed vessels killed cholera microbes, thus preventing the spread of cholera epidemics in humans. Hankin did not identify the specific factor that was responsible for this phenomenon, but defined it as a heat labile antiseptic substance passing through the pores of fine porcelain filters [63,64]. However, the discovery of bacteriophages is attributed to two other scientists, Frederick Twort, an English bacteriologist, and Felix d’Herelle, a French-Canadian microbiologist, working independently [65,66,67].

In 1915, 20 years after Hankin’s observation, Twort, for the first time, hypothesized that the abovementioned substance might be an ultramicroscopic virus, but he did not pursue his discovery. In 1915, d’Herelle began an investigation of an outbreak of severe dysentery hemorrhage among French soldiers that were stationed in Maisons-Laffitte, which contributed to the discovery of bacteriophages and the subsequent treatment of dysentery with phages in humans. Although he apparently first discovered the phage phenomenon in 1910, examining microbiological methods of controlling locust epizootic in Mexico. On 18 October 1916, d’Herelle proposed the name “bacteriophages” (formed from two words i.e., “bacteria” and “phagein”) for the viruses eating bacteria. The results of his dysentery studies were demonstrated during the September 1917 meeting of the Academy of Sciences, and then published in the meeting’s proceedings [65,66].

In 1919, therapeutic studies that were based on d’Herelle’s antidysentery phage were conducted at Hospital des Enfants-Malades in Paris. This was probably the first attempt to use phage therapy in the treatment procedure of bacterial infections in humans [68]. d’Herelle is officially considered a discoverer of bacteriophages due to actively conducted phage research.

## 5. Phage Preparations and Therapy of Bacterial Infections

After d’Herelle’s investigations that were associated with effective treatment of dysentery, interest in the phage therapy significantly increased and the production of phage preparations began. However, the first reported administration of lytic bacteriophages in the treatment of human infectious diseases did not appear until 1921, when Richard Bruynoghe and Joseph Maisin used phages to treat staphylococcal skin infections [69]. Subsequently, the commercial laboratory of d’Herelle in Paris started producing phage preparations against various bacterial infections (e.g., Bacte-coli-fage, Bacte-pyo-fage, Bacte-intesti-fage, Bacte-staphy-fage and Bacte-rhinophage). Phage preparations were also made in the USA. In the 1940s, the Indianapolis company of Eli Lilly produced seven therapeutic phage preparations in the form of phage lysates (e.g., Neiso-lysate, Staphylo-lysate, Colo-lysate, Ento-lysate) or preparations in a water-soluble jelly-base (e.g., Staphylo-jel, Colo-jel, Ento-jel) targeted against streptococci, staphylococci, *Escherichia coli*, and other bacterial strains. However, the safety of phage preparations that were used to treat or control bacterial infections and diseases in humans was very controversial. In addition, the discovery and commercial production of antibiotics in the 20th century (beginning with the discovery of penicillin in 1928) to fight various pathogens, such as bacteria and fungi, reduced phage production in many Western countries [70,71]. Henceforth, phages were mainly used to treat infections in Eastern Europe, such as Georgia, Russia (countries of the former Soviet Union), and Poland. In these countries, the following institutions were responsible for the production of phage preparations i.e., the Eliava Institute of Bacteriophage, Microbiology, and Virology (EIBMV) of the Georgian Academy of Sciences, Tbilisi, Georgia, and the Hirszfeld Institute of Immunology and Experimental Therapy (HIIET) of the Polish Academy of Sciences, Wroclaw, Poland [64].

The problem of emergence of bacterial pathogens resistant to many drugs and chemotherapeutic agents began due to the excessive use of currently available antibiotics for the treatment of various diseases since the 1930s. Therefore, in 2016, World Health Organization (WHO) created a priority list of antibiotic-resistant bacteria to support studies and development of new drugs against resistant strains. According to WHO analysis, 60% of pathogenic bacteria, including the *Enterobacteriaceae* family, acquired resistance to commonly used antibiotics, including recently isolated carbapanems and third generation cephalosporins. However, given the above data, it can be assumed that all pathogens will acquire 100% resistance over the next few years [72,73].

The development of diverse antibiotic resistance mechanisms (i.e., multidrug and extensively drug-resistance) by bacterial strains is a huge economic burden for many countries, associated with the treatment of infections in humans. In addition, the spread of drug-resistant pathogens poses a serious threat to human morbidity and related mortality worldwide. In this regard, research is very important to look for and develop alternative treatments for different diseases that are caused by microorganisms. In recent years, interest in phages as antimicrobial agents has been renewed. Therefore, phage therapy that is based on the lytic phages seems to be the best way to solve the problem of drug resistance among bacteria (including uropathogenic strains responsible for UTIs), as well as limiting drug abuse and their misuse in the therapeutic profile. Animal studies also emphasize the efficacy of phages as antimicrobial agents used for the treatment of single and complex bacterial infections [74,75,76,77]. It was estimated that phages enter the blood of laboratory animals within 2–4 h, and then, along with the bloodstream, are transported to internal organs (e.g., kidney, spleen, liver). The tests carried out also indicate the persistence of administered phages. The phages can remain in the human body for up to several days, after a single oral dose. Additionally, when the bacteria are killed, the phage titer decreases until they are completely removed from the body (in urine or feces). No differences in pathogenesis or potential toxic side effects were observed after phage therapy, particularly when compared to antibiotic treatment, based on animal models. Furthermore, phages did not affect normal microflora and did not worsen dysbiotic disorders in the animals tested. Phage therapy was efficient and safe. For these reasons, bacteriophages can be an alternative therapy in living hosts selectively destroying pathogenic agents. [64,78,79].

## 6. Phage Types and the Potential of Phage Therapy in the Treatment of UTIs Caused by UPECs

Lytic therapeutic phages as bactericidal agents kill target bacteria through intracellular replication and lysis. The lysis of target host cells is a complex cascade that involves structural and regulatory phage proteins [64]. In the treatment of urinary infections, which are mainly caused by UPECs, natural lytic phages alone (single receptor systems) or in the form of cocktails (dual-receptor systems), genetically modified or engineered phages, phage enzymes (native or modified), as well as a combination of phages and antibiotics (synergistic mechanism) represent the innovative and promising therapeutic alternatives (Table 3) [8,39,40,58,64].

### 6.1. Mono- and Polyphages, Phage Cocktails

Given the number and type of phages used in UTI therapeutic preparations, monophage and polyphage therapy can be distinguished [64,80,81]. Monophage therapy is based on the use of one type of phage, which narrows host range during treatment procedure. In addition, this therapy might be associated with the emergence of bacterial resistance to a particular type of phage or stimulation of immune response leading to the production of anti-phage antibodies during long-term therapy. In turn, polyphage therapy involves the use of more than two phages or phage cocktails with broader host specificities, which is a condition to overcome bacterial resistance to phages and host immune function. This therapeutic approach defines the targeting of various bacterial strains or multiple species that are responsible for UTIs by phage recognition of different receptors on the surface of urothelial cells. Such therapy is effective, even when one of the bacterial receptors undergoes mutation and, above all, enables the treatment of UTIs related to biofilm formation (a complex consortium of bacteria) by UPECs. The development of biofilm makes the treatment of such infections difficult due to bacterial resistance to antibiotics and the host’s immune system to eliminate them. However, the biofilm structure is susceptible to the action of bacteriophages that are contained in phage cocktails or phage depolymerase enzymes, which have the ability to degrade the biofilm that was established by various uropathogenic strains [8,81,82].

Research performed indicated that monotherapy alone or in combination with antibiotics can be used to treat acute and chronic inflammatory urologic diseases that are caused by *E. coli*, *Staphylococcus,* and *Proteus* species. During the tests, phage preparations were locally and orally applied in 46 patients. The effectiveness of phage treatment was estimated at 92% (observed as a significant clinical improvement) and bacteriological clearance 84% [83].

In turn, other studies showed that phage cocktails can be more effective in the treatment of UTIs than the use of individual phages. The first phage cocktail targeting MDR UPECs was composed of T4 phage and KEP10 phage, which were isolated from wastewater. The above cocktail was introduced into the peritoneal cavity of the mouse, and then through the blood that was transferred to its internal organs [39]. The combination of these two phages in the form of a cocktail are the first therapeutic candidate for phage therapy used in the treatment of UTIs caused by UPECs.

UPECs have the capacity to adhere, colonize, and invade epithelial cells i.e., processes that represent the main stages during the pathogenesis of UTI [4]. The invasion allows for bacteria to acquire a cellular niche within the urinary tract, where they can survive in a protected manner, especially under the influence of antibiotics. It is known that *E. coli* attached to urothelium can be up to 100 times more tolerant to antibiotics. In this context, the T1-, T4-, and phiX174-like phages were used to reduce the loads of UPEC adhered to epithelium and evaluate their efficacy after the infection of bacteria, under static and shaken conditions. Among them, the T1 phage was most effective in killing bacteria attached to the urothelium, causing a 45% reduction of bacteria after two hours of treatment. This phage was also characterized by a wide lytic spectrum against clinical *E. coli* isolates and the ability to infect different antibiotic resistant strains [60].

### 6.2. Single- and Dual-Receptor Phages, Polysaccharide Depolymerases

The recognition of bacterial receptors by phages is necessary for initiating the infection cycle and lysis of the host cells. In the context of UPECs, the main phage receptors include the components of surface-exposed pili, fimbriae (i.e., homo- or heteropolymeric adhesive structures), or afimbrial bacterial sheaths, O-antigen of lipopolysaccharides (LPSs), oligosaccharides of the outer and inner core of LPS, and flagella that are attached to the cell wall. Mutational or conformational changes of these receptors can become a source of bacterial resistance to the appropriate phages used in the given therapy [2,80,84,85]. When considering the number of bacterial receptors recognized by phages (determining their host range), they can be divided into two groups, as single-receptor phages and dual-receptor phages. The dual phages, like phage cocktails, have the ability to recognize more than one host receptor and infect several uropathogenic strains. In addition, they may reduce the appearance of phage-resistant UPECs mutants. Phage therapy targeting different receptors can be very important and valuable in the treatment of UTIs. An excellent example of the effectiveness of the abovementioned therapeutic strategy is the use of a combination of two phages SP21 and SP22, binding to various host receptors (i.e., an outer membrane protein, OmpC, and LPS, respectively) of enterohemorrhagic *E. coli* O157:H7 strain. The simultaneous use of SP phages resulted in a significant increase in the time of emergence of phage-resistant *E. coli* (up to 30 h) as compared to using each of these phages separately [86].

Individual phages that recognize more than one bacterial receptor are also known. This group includes, for example, the T4 and T2 *Enterobacteriaceae* phages belonging to the *Myoviridae* family. The phages bind to two different receptors (LPS and OmpC or a surface protein and bacterial polysaccharide, in the case of T4 and T2 phages, respectively) located on the surface of bacterial host cells [87,88,89].

Very interesting and therapeutically valuable are phages that encode two different enzymes, functioning as polysaccharide depolymerases (PDs) discovered over 60 years ago [90]. The enzymes are responsible for the recognition and depolymeryzation of capsular and structural polysaccharides, including exopolysaccharides (EPS). The polysaccharide component is secreted by bacteria into the surrounding environment and represents the dominant element of existing biofilms (almost 90% of all biomass). PDs may exist as a free form that is dispersed in the culture medium or attached to a phage tail. This heterogeneous group of enzymes includes hydrolases (glycanases) or polysaccharide lyases [91,92]. The large-scale enzymes also represent potential agents for combating infections that are associated with the development of biofilm and against capsular bacteria that cause severe infections, such as sepsis, meningitis, pneumonia, osteomyelitis, septic arthritis, and pyelonephritis [93]. To date, many PDs have been found in phages that infect *E. coli* strains [94,95,96,97,98]. An example of such phages is phage named φK1-5, which is a member of the *Podoviridae* family. The φK1-5 phage encodes a K5 lyase and endosialidase, with the enzymes belonging to capsule specific tail fiber proteins that allow for the hydrolysis of polysaccharide capsules (i.e., K antigens) and infection on both the K1 and K5 different strains of *E. coli* [97]. Other examples of DP-producing phages are coliphage φK5 and φK20, which use K5 and K20 polysaccharides as major receptors. The φK5 is capable of disrupting capsular polysaccharides that are produced by *E. coli* K5 or K95. In contrast, the φK20 phage has enzymatic degrading activity towards both *E. coli* K20 and K5 antigens [96,99,100]. Capsule-free bacteria have lower virulence and are more susceptible to the immune system (including phagocytosis of peritoneal macrophages), as demonstrated in an animal model that is based on neonatal rats. A single dose of phage endosialidase E, endoE (administered 24 h after the onset of bacterial infection), protected animals undergoing experimental systemic *E. coli* K1 infection and reduced their mortality rate within seven days of testing from 80–100% to 0–10%, respectively [101].

### 6.3. Engineered and Genetically Modified Phages

In addition to host-specific phages, engineered or genetically modified phages with desirable properties represent a new strategy in the treatment of UTIs, caused, in particular, by antibiotic-resistant uropathogens. These phages are obtained while using genetic methods, genetic engineering, and other technologies e.g., CRISPR-Cas (Clustered Regularly-Interspaced Short Palindromic Repeats-Cas) genome editing system, homologous recombination, in vivo recombineering, phage recombineering of electroporated DNA, rebuilding phage genomes in vitro, whole-genome synthesis, or assembly from synthetic oligonucleotide. Modified bacterial viruses can be applied to detect and control bacterial infections in humans. They can also be a source of antibacterial agents or serve as carriers for the delivery of therapeutic genes, drugs, or vaccines [102,103].

A valuable example of the abovementioned therapy are genetically modified phages that can be used to kill intracellular bacteria in human urinary cells. Based on the CRISPR-Cas system and homologous recombination, bacteriophage K1F infecting *E. coli* K1 was genetically modified by the insertion of a green fluorescent gene into its genome to obtain the fluorescent phage K1F-GFP. The strain of *E. coli* K1 is a nosocomial pathogen that is responsible for urinary tract infections, neonatal meningitis, and sepsis. The above phage and *E. coli* EV36-RFP (a K-12/K1 hybrid displaying the K1 capsule) were able to invade epithelial cells of the urinary bladder T24 transitional cell line by phagocytosis. Inside the T24 cells, bacteria were killed by engineered phage K1-GFP. Additionally, the viruses and bacteria were degraded by LC3-associated phagocytosis and autophagy. These types of phages in the future may be a valuable therapeutic tool in the treatment of infections that are caused by *E. coli* strains, as well as other uropathogens, including MDR strains [104].

All of these studies indicate the efficacy of phages as an alternative and innovative treatment for UTIs that are caused by UPECs and highlight the need for further research to develop and validate these therapeutic methods.

## 7. Bacteriophages in the Treatment of Urologic Diseases Associated with Catheterization and Biofilm Formation by UPECs

UPECs are one of the main causative factors of complicated urinary infections, which mainly affect people undergoing indwelling catheterization, e.g., in hospitals or social care homes. CAUTIs may occur as both symptomatic and asymptomatic infections, equally affecting men and women (especially elderly, persons with diabetes and long-term catheters) [105]. The time of catheterization generally increases the risk of bacterial infection (CAUTI risk of 3–7% within one day of catheterization) and it is related to the bacterial detection in the urine (as bacteriuria). The more frequent asymptomatic form of CAUTI can cause very serious complications, including bacteraemia, urosepsis, and even death [18,105]. Catheterization is associated with the adhesion of bacteria to the surface of the catheter and epithelial cells, involving surface-exposed adhesive structures, such as pili or fimbriae. Ultimately, bacteria form organized complex cellular communities that are embedded in a self-produced extracellular polymer, EPS i.e., a biofilm that is a huge clinical problem in many countries worldwide [91,106]. Therapeutic difficulties of CAUTIs can be associated with the multispecies structure of a biofilm, which causes its resistance to e.g., a given antibiotic, phage preparation, or depolymerase enzyme.

### 7.1. Strategies of Phage Therapy to Combat Biofilm

The capability of UPEC strains to form biofilms further complicates UTI treatment with antibiotics and results in the emergence of drug resistant strains. This is due to the limited diffusion of antibiotics by EPS or the presence of bacterial enzymes degrading bactericidal factors. Laboratory tests highlight that the biofilm structures can be over 1000 times more resistant to antibiotics than planktonic bacteria. Bacterial biofilms are also responsible for many chronic and persistent infections, which is associated with increased bacterial resistance to drugs, phagocytosis, and other elements of the host’s defense system [106]. However, the formation of biofilms by UPECs does not protect bacterial cells from phage killing. There are several potential methods for biofilm eradication. The first approach preventing biofilm formation on biomaterials is based on coating the catheters with an inert hydrogel, which is then covered with phages. The second strategy refers to controlling an existing biofilm by treating it with a single phage or phage cocktails. Finally, the third assumption (extensively studied) is associated with the use of anti-biofilm agents representing a new group of antimicrobials i.e., natural lytic phages or engineered phages producing PDs or phage-derived enzymes, as well as combinations of phages or PDs with antibiotics or disinfectants [8,93,107].

### 7.2. Therapy Based on Selected Doses of Phages and Phage Cocktails

In the biofilm, the phages move through water channels, and PDs dissolve the biofilm matrix, which allows for the phages to reach all biofilm layers containing bacterial aggregates [92,107,108]. It is assumed that one dose of phage is sufficient to eradicate biofilm. Researches demonstrated the ability of bacteriophages to remove mature biofilm that was formed by *E. coli* and *Proteus mirabilis* (responsible for CAUTIs or infections of indwelling urological devices) and to prevent its formation on urological catheters [109]. Other studies related to the action of bacteriophages on bacterial biofilm have shown that properly selected doses of phage with appropriate PD are necessary for effective phage therapy. Accordingly, it has been demonstrated that a single dose of a Shiga-toxin encoding phage H-19B can rapidly lead to the almost complete lysogenisation of an existing mature *E. coli* biofilm, followed by continuous release of infectious H-19B particles. For comparison, a single dose of an obligatory lytic phage T7 quickly led to the emergence of bacterial resistance in the biofilm structured consortium [110]. Another example are experiments that are based on the coevolution model of *E. coli* and the lytic RNA bacteriophage (Qβ) in a spatially unstructured environment, which allowed for them to co-propagate for 54 days. In this biological model, the phage represents a parasite with a smaller genome size (4217 bases) and a higher spontaneous mutation rate (i.e., 1.5 × 10^−3^–10^−5^ per base per replication). In contrast, bacterium as a phage host has a larger genome size (4.64 Mbp) and lower spontaneous mutation rate (i.e., 5.4 × 10^−10^ per base pair per replication). During the test, QB phage and *E. coli* were both able to change their phenotypes to coexist in an arm race. The studies highlight the possibility of the appearance of phage-resistant bacteria and phage variants that are capable of infecting resistant strains, from which further mutants resistant to them may arise by using continuous adaptation and counter-adaptation mechanisms [111].

Good results in biofilm eradication were also obtained while using phage cocktails. Recent studies allowed for the isolation of phages (i.e., vB_EcoP_ACG-C91, vB_EcoM_ACG-C40, and vB_EcoS_ACG-M12) that were effective against UPECs and capable of destroying biofilms (developed by the bacterial populations on polystyrene microtiter plates). The phages caused lysis of 80.5% among the UPEC isolates studied. These phage preparations can have great potential in the treatment of UTI that is caused by UPEC forming biofilms on the medical devices or intracellular bacterial communities (IBCs) within the urinary tract [82].

### 7.3. Therapy Based on Phage/PD-Antibiotic or Disinfectant Combinations

In many cases, therapies that are associated with combinations of phages or PDs with antibiotics or disinfectants, such as chlorine compounds or iron antagonists, seem to be more effective in treating biofilm-related infections than those based on phage preparations alone. In combination therapy, phages condition the diffusion of antibiotics into the deeper layers of the biofilm structure, thus obtaining a bactericidal concentration, unattainable with antibiotic therapy alone [112]. Detailed research revealed that sub-lethal concentrations of some antibiotics can significantly stimulate the production of some virulent phages by the virulent bacterial cells. This phenomenon has been defined as Phage-Antibiotic Synergy (PAS). The PAS effect was observed in several independent host-phage systems. The first concerned the production of phage PhiMFP by infected UPEC (MFP-*E. coli* MFP system) that was cultured in the presence of cefotaxime, a cephalosporin. A low dosage of cefotaxime (20 ng/mL) caused a seven-fold increase in the number of phages released during bacterial lysis as compared to the number of phages obtained after infection of bacteria that grew on agar plates without the addition of a suitable antibiotic. A similar effect was noticed in other systems involving the T4-like phages, with beta-lactam and quinolone antibiotics and mitomycin C [113]. Further research revealed that an increase in sub-lethal cefotaxime concentrations was the cause of the increase in T4 plaque size (determined by plaque diameter) and T4 concentration. In addition, the simultaneous application of T4 phage and cefotaxime resulted in more effective in vitro destruction of the T4 host *E. coli* ATCC 11303 biofilms when compared to the treatment of these complex structures with the antibiotic alone. It was also observed that the T4 phage titers of 10^4^ PFU/mL and 10^7^ PFU/mL reduced the minimum concentration of cefotoxime that is needed for the eradication of bacterial biofilm from 256 mg/mL to 128 and 32 mg/mL, respectively [114]. This is the first study demonstrating the important role of synergistic phage-antibiotic combinations being used to control *Escherichia coli* biofilm in vitro. Combination therapy, in addition to UTI therapy, also significantly reduces the emergence of resistant bacterial variants when compared to methods that are based on the use of phages alone [112].

Another form of combination therapy is based on the use of phages and factors that limit bacterial access to nutrients that are necessary for their growth and biofilm formation, e.g., iron. Iron ions are responsible for the transition of planktonic bacteria to settled forms and the stabilization of polysaccharides in the biofilm structure. It is also known that biofilm-associated urinary tract-infectious *E. coli* strains are particularly dependent on iron. In this case, biofilm formation by *E. coli* was inhibited by the addition of divalent metal ions, such as Co(II) and Zn(II) to the culture medium. These ions have a greater affinity for the main controller protein of iron uptake (Fur) as compared to iron. A reduction in biofilm development by UPEC was analyzed in microtiter plates, flow chambers, and on urinary catheters. These studies indicate the possibility of using the iron uptake system as a target to control and eliminate bacterial biofilm [113].

### 7.4. Therapy Based on Engineered Phages

It is necessary to search for new forms of phage therapy using genetic engineering methods due to the fact that bacterial biofilms are crucial in the pathogenesis of many bacterial infections and are difficult to remove while using traditional antimicrobial treatment. Phages modified in this way will contain genes that give them defined properties. An example of such a system is engineered enzymatic phage T7_DspB_, which expresses EPS-degrading enzyme dispersin B (DspB) that is naturally produced by *Actinobacillus actinomycetemcomitans*. This enzyme hydrolyses β-1,6-*N*-acetyl-d-glucosamine, an adhesin that is necessary for the development of biofilm by the laboratory *E. coli* K-12 and clinical *E. coli* strains. The T7DspB phage was found to be more effective at destroying biofilm than T7 phage. The above research highlights the wide possibilities of using engineering phages in eliminating biofilm in medicine and industry (during UTIs and technological processes) [115].

Intensive research indicates the high efficiency of abovementioned strategies in killing bacteria developing biofilm when compared to traditional antibiotic-based therapy. Therefore, phage-based therapy can be an alternative and complementary form of treating biofilm-related infections, including UTIs [116].

## 8. Clinical Trials as a Condition for Safe Phage-Based Therapy Against UTIs and Gastrointestinal Diseases—Selected Examples

Many phages have been analyzed for their potency as antibacterial agents used in phage therapy to control and treat UTIs and gastrointestinal diseases. However, only several of them have been clinically tested, while taking the different clinical stages into account. The performed studies indicate the safety and efficiency of phages in the treatment procedure of UTIs and diarrheal diseases that are caused by *E. coli* and other bacterial strains. Despite the high effectiveness of phage therapy, the number of patients studied is still low and there is also a deficiency of randomized clinical trials (RCTs) [40,117,118,119]. Here, we briefly reviewed clinical studies related to the use of phages in the therapy of UTIs and diarrheal diseases.

The important approach to clinical studies associated with uropathogenic strains (i.e., *Staphylococcus aureus*, *E. coli*, *Streptococcus* spp., *Pseudomonas aeruginosa*, *Proteus mirabilis*) was based on the commercially available pyophage (Pyo) (Eliava BioPreparations Ltd., Tbilisi, Georgia) [40]. The research was preceded by the design of a prospective, randomized, placebo-controlled, double-blind clinical trial to assess the safety and efficacy of intravesical phage therapy in humans [120]. The investigations consisted of two phases and involved phage adaptation followed by treatment of patients who were to undergo transurethral prostate resection. During the first phase, patients with positive urine culture were selected. Subsequently, based on in vitro analyzes, phage sensitivity to isolated uropathogens was determined, which was 41% and 75%, after phage adaptation cycles, respectively. The second phase included the treatment of patient with adapted Pyo bacteriophage in a non-blinded fashion (in vivo pilot series). The therapy was effective in six out of nine examined patients (~67%). A reduction in bacterial titer was observed in these patients without any adverse effects [40]. The results of the above prospective two-phase study underline the high efficacy and safety of adapted phage therapy in the treatment of UTIs. Therefore, well-designed RCTs are highly justified in order to determine the role of this alternative form of therapy in controlling infections that are caused by UPEC and other bacterial strains in association with UTIs. Subsequently, the clinical trial design (phase II/III) based on the pyophage cocktail useful in the treatment of UTIs was published by EIBMV, Georgia [121].

In addition to clinical trials, the preclinical studies of polyphage therapy, including two novel virulent phages (i.e., the podovirus vB_PmiP_5460 and the myovirus vB_PmiM_5461), also seem to be very promising to fight catheter-related UTIs, caused by *Proteus mirabilis.* A significant decrease in *P. mirabilis* biofilm development on catheters was obtained (up to 168 h of catherization) based on the dynamic biofilm model simulating CAUTIs [122]. CAUTIs are very difficult to treat due to the high tolerance of biofilm structures to the antibiotics used. However, the above phages can be a useful therapeutic tool for controlling and preventing catheter-associated biofilm formation by *P. mirabilis*.

Oral phage clinical studies have also been performed, based on T4-like phage preparation targeted against EPEC and ETEC *E. coli* strains, which are often responsible for diarrheal infections. This phage cocktail was evaluated in a phase I placebo-controlled study in healthy adult volunteers from Switzerland [123] in 2005 and Bangladesh in 2012 [124]. After two days of oral phage administration (in drinking water), no adverse effects were observed in the subjects (analyzed by physical examinations or laboratory tests of hepatitic, renal, and hematological functions). However, the phages did not proliferate in the intestine at a significant level and did not reduce the total *E. coli* in feces. Only small doses of phages were recovered in stool samples. In addition, no phages and antibodies directed against them (i.e., T4-specific neutralizing antibodies) were found in the blood of the tested individuals [16,107]. The above studies represent the first tests to assess the safety and bioavailability of oral phages use in humans. These are also preliminary analyses regarding the usefulness of phages in the treatment of diarrheal diseases.

Another example and continuation of the aforementioned studies is an attempt to apply and compare the effects of two different oral phage cocktails, a T4-like phage cocktail, and a commercial Microgen ColiProteus phage cocktail (targeting *Escherichia coli* and *Proteus* infections, Russia) to treat *E. coli* acute diarrhea in 6–24-month-old male boys from Bangladesh [125]. This approach was a placebo-controlled, double-blind trial, investigating the safety and effectiveness of selected phage cocktails in children. The phages were transported into the intestine and they were well tolerated with no adverse effects (observed clinically and using clinical chemistry), but did not improve diarrhea in the tested subjects. In addition, the detection of phages in the stools was dependent on the oral dose used. Furthermore, no significant difference between the placebo and phage treatment group was observed. The therapy was characterized by low efficiency, probably due to low *E. coli* titers, and the contribution of other pathogens, such as *Streptococcus* spp., as diarrhea etiologic agents. Gastric buffering prior to phage administration could be another reason for the low efficacy of the therapy. In addition, lowering the pH in the stomach caused a reduction in the number of surviving phages that did not have optimal conditions for amplification. This required the use of high oral doses of phage preparations in a placebo-controlled randomized phase I safety trial. The tests were terminated for the following reasons. More research is needed in order to assess the usefulness of the above phage preparations in the treatment of individuals suffering from acute diarrhea episodes [125,126].

The Bacteriophage for Gastrointestinal Health (PHAGE) study represents different approach of phage clinical trial based on a phase I randomized, double-blind and placebo-controlled crossover intervention. The purpose of this study was to assess the safety and tolerability of a 28-day supplemental oral consumption of a mixture of 4 bacteriophages, LH01-Myoviridae, LL5-Siphoviridae, T4D-Myoviridae, and LL12-Myoviridae (as the potential prebiotics) in adults with mild to moderate gastrointestinal complaints. The efficacy of the procedure used was comparable to placebo in reducing small intestinal pain and more effective than placebo in reducing the symptoms of colon pain or abnormal stomach function. In addition, the treatment was unsuccessful in reducing perceived gastrointestinal inflammation [127,128]. However, these are the first studies focusing on the use of phages as probiotics, regulating the functioning of the gastrointestinal system.

So far, phage therapy has not been validated. However, compensatory phage therapy is currently used in patients for whom classical antibiotic treatment was ineffective or inapplicable. There are also cases of humans suffering from chronic, resistant, and life-threatening infections who go to foreign bacteriophage centers (in the countries of Central and Eastern Europe) to try alternative phage therapy (including all ethical and legal issues). In this way, medical tourism for phage therapy develops [117,129]. At the end of January 2018, the Belgian Federal Government developed regulations regarding the production of phages and the clinical use of phage therapy. An innovative approach, so-called “magistral phage preparation” (or “compound prescription drug preparation” in the US), was also introduced in Belgium, based on which, according to the doctor’s prescription, non-standard phages (used for the therapy of individual subjects) are prepared in the laboratory [130]. A similar therapeutic approach, known as experimental therapy, has long been used in Poland at HIIET [131,132]. However, extended clinical research is still needed to make phage therapy a routine and widespread therapeutic form, not just experimental treatment.

## 9. General Conclusions

UTIs are one of the most common and widespread microbial diseases affecting millions of people every year around the world. UTIs are often associated with recurrences, long-term chronicity, and repeated antibiotic treatment. It is estimated that approximately 50% of women experience symptomatic urinary tract infection at least once in their lifetime, which requires antibiotic therapy [11,109,133]. In the USA alone, UTIs are responsible for 11-million medical appointments and 100,000 hospital admissions each year [1].

The major causative agent of complicated and uncomplicated UTIs are UPECs. The *E. coli* isolates are the most common cause of nosocomial infections often associated with the development of MDR or XDR uropathogenic variants. The emergence and rapid spread of antibiotic resistant UPECs is a serious therapeutic problem due to the limitation of the use of many common pharmaceuticals in the treatment of UTIs [53,134].

The prevalence of bacterial multi-drug resistance is a serious medical problem that requires the development of new pharmacological agents and alternative forms of therapy, such as vaccines, receptor analogues, pilicides, or phagotherapy [134]. In the last decade, phage therapy based on bacterial viruses aroused the interest of many scientists from around the world, not only Russia, Georgia, and Poland, but also many Western countries, as an innovative method of treating different bacterial-related infections and often as the only life-saving alternative [2,8,60,82,135].

Various forms of bacteriophages can be used in UTI therapy, including natural lytic phages, phage cocktails, genetically modified and engineering phages, phage lytic enzymes and their derivatives, and also phage-antibiotic combinations that are based on the synergistic mechanism. Bacteriophages as a potential therapeutic tool against UTIs have many advantages when compared to routinely used antibiotic treatment (including their host range, application rate, replication mechanism, mode of action, production costs, biofilm eradication, and other) [8,64]. Although phage therapy is characterized by high effectiveness and safety in the treatment of UTIs, the number of clinical trials performed (phase I, I/II, II, and II/III) is low, and there are also no randomized control trials (RCTs). However, several completed clinical trials for the treatment of UTIs and diarrhea in children (caused by UPEC, EPEC, ETEC, and *P. aeruginosa*) while using phage cocktails and lytic enzymes confirmed their efficacy, safety (no adverse effects in studied patients), therapeutic utility, and alternative to routinely used antibiotics [40,121,136].

The use of phages in the treatment of UTIs that are caused by UPECs and other uropathogenic strains still requires extended clinical research in order to validate this method of therapy. Studies carried out so far clearly indicate the wide possibilities of using different forms of phages and their safety to control UTIs and eliminate uropathogenic bacteria (including UPECs) and the biofilm developed by them in the urinary tract and on medical devices (especially on catheters).

Undoubtedly, in the future, phages as a kind of pharmaceutical preparations will be a valuable alternative to commonly used antibiotics to treat UTIs, caused by uropathogenic strains, in particular MDR and XDR *E. coli* clinical isolates.

## Figures and Tables

**Table 1 antibiotics-09-00304-t001:** Clinical types of urinary tract infections (UTIs) [4,14,15,16,17,18].

Uncomplicated UTIs	Complicated UTIs
PREDISPOSITION AND OCCUERENCE	
**Healthy Individuals:** not pregnant not catheterized without urinary tract disorders	**Individuals with:** neurological disease immunosuppression, immune disorder renal failure or transplantation calcui, indwelling catheters, drainage devices Pregnant women

**Table 2 antibiotics-09-00304-t002:** Pathotypes of *E. coli* strains [20,21,22].

Intestinal/Diarrheagenic *E. coli*	Extraintestinal *E. coli*
MAJOR PATHOGENIC CATEGORIES	
**InPEC Pathotypes**: enteropathogenic *E. coli* (EPEC) enterohemorrhagic *E. coli* (EHEC) Shigatoxin producing (STEC) enterotoxigenic *E. coli* (ETEC) enteroinvasive *E. coli* (EIEC) enteroaggregative *E. coli* (EAEC) diffusely adherent *E. coli* (DAEC)	**ExPEC Pathotypes**: uropathogenic *E. coli* (UPEC) neonatal meningitis *E. coli* (NMEC)
**Factors Inducing Pathogenicity:** immune disorder, immunodeficiency crossing gastrointestinal barriers (e.g., peritonitis) evolutionary modification of the bacterial genome	

**Table 3 antibiotics-09-00304-t003:** Types of phage therapy against Urinary tract infections (UTIs) caused by uropathogenic *E. coli* strain (UPEC) and other uropathogens [8,58,64,80,81].

Type of Therapy	Characterization
Monophage Therapy	Single phage type Narrow host range Emergence of bacterial resistance Minimal damage of the natural host flora Infection of one type of bacterial strain or a species
Polyphage Therapy or Phage Cocktails	Two or more phage types Broader host range and different host specificity Overcoming bacterial resistance Treatment of biofilm-related infections
Single-Receptor Phages	Single phage type Binding to a specific host receptor Targeting of a specific bacterial species
Double-Receptor Phages	Single or two phages Binding of various host receptors Action against several types of different strains of bacteria
Polysaccharide Depolymerases (PDs)	Two groups of enzymes: hydrolases (glycanases) and polysaccharide lyases Recognition and depolymerization of capsular and structural polysaccharides (e.g., EPS) Infection of single or different strains of bacteria Action against encapsulated infectious bacteria Treatment of biofim-related and severe infections e.g., sepsis, meningitis, pneumonia, osteomyelitis, septic arthritis and pyelonephritis
Engineered and Genetically Modified Phages	Phages with desirable properties Potential carriers for the delivery of therapeutic genes (enzymes), drugs and vaccines Treatment of infections caused by different uropathogens also MDR strains
Phage/PD-Antibiotic or Disinfectant Combinations (e.g., iron antagonists)	Synergistic mechanism based on sub-lethal concentration of certain antibiotics Reduction of bacterial growth by targeting the iron uptake system Overcoming bacterial resistance Treatment of biofilm-forming infections

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
