# Peer review of "Phage Therapy as a Novel Strategy in the Treatment of Urinary Tract Infections Caused by E. Coli"

_antibiotics, 2020, doi:10.3390/antibiotics9060304_

Round 1

Reviewer 1 Report

This manuscript gives a good overview of the use of phage therapy and makes a valid argument that this strategy be considered as a stand-alone or combination treatment for UTIs.  The clinical need is stated well.  In reading the text, details of these clinical studies are missing and since phage therapy is being critically reexamined, it would be beneficial to describe this work in more detail.  There are a small number of recent reviews that address some clinical trials (Kortright et al., Cell Host Microbe 25(2): 219-232, 2019; Gordillo Altamirano and Barr Clin. Microbiol. Rev. 32(2): e00066-18, 2019; and Gorski et al., Viruses 10(6):288, 2018). Overall, several comments are listed with the intention of enhancing the caliber of the manuscript.

  1. In section 1.1, the authors describe some of the virulence mechanisms involved in UTIs. While there have been a number of classical studies done in this aspect, there is now a growing field of genome and transcriptome-based information related to uropathogenic virulence (labs of HLT Mobley, SJ Hultgren, and others). Some of these studies identify notable changes between intestinal and uropathogenic E. coli.

  1. Another consideration would be to include a mention of the urinary virome (e.g. Miller-Ensminger et al., J. Bacteriol. 200(7): e00738-17, 2018; Santiago-Rodriguez et al., Front. Microbiol. 6: 14, 2015). There are a number of bacteriophage in the urinary tract and one could argue that co-evolution of uropathogens and phage would occur in the urinary tract, which could result in the generation of phage resistance.

  1. Line 60 typo Enterobacteriaceae is the correct spelling.

  1. Line 230 typo in section 6.1 heading - cocktails is the correct spelling

  1. In lines 242-245, the authors make the argument that the biofilm structure is susceptible to phage. I would recommend mentioning depolymerase enzymes as a mechanism as they are mentioned elsewhere in the manuscript.

  1. One issue related to many infections including some UTI’s is the polymicrobial nature. Certainly, this occurs in CAUTIs and may also be an issue in acute or recurring cystitis. In a biofilm situation, polysaccharides produced by one biofilm population may not be susceptible to phage depolymerase enzymes targeting another organism.

  1. The authors are proposing the use of clinical trials. While the number of clinical trials related to UTIs may be low, a number have been conducted in other infections. It would be quite beneficial to describe these and also present some of the data rather than simply mention the conclusions of the previous investigators. As part of the presentation, were any undesirable side-effects seen with the patients involved?

  1. In a similar comment about animal experimentation, were any differences seen in pathogenesis or potential toxic side effects (if any) from phage therapy, particularly when compared to antibiotic therapy?

Reviewer 2 Report

The manuscript presents a review on the use phage therapy for the treatment of urogenital infections caused by E. coli. It is an introductory review of the theme with only around 30% of the references used published in the last five years and around 20% of the references with more twenty years old.

In my opinion the manuscript may be published after the after some minor clarifications/corrections:

In my opinion the title is too generic as the review only focuses the use of phages in the treatment of E. coli infections. Although, E. coli is the main cause urinary tract infections (85% according to the authors), phages have also been studied in other causing microorganisms (for instance Klebsilla).

The reviews mainly mention the use of phage therapy in preclinical models. Some information or discussion of eventual clinical studies could enrich the review.

Some typos:

Line 60 (Enterobacteriaceae) and in Table 2

Round 2

Reviewer 1 Report

I applaud the authors for addressing my prior concerns, and also those of the other reviewer. Providing additional insight into phage therapy, describing some outstanding work on intestinal and uropathogenic E. coli by several major labs, and acknowledging current work on the virome of the urinary tract is quite helpful. I consider this manuscript in its present form to be acceptable.

Having made this recommendation, given the prolonged history of bacteria and phage and their associated co-evolution; some measure of phage resistance is very likely and should be continuously monitored.